# Study on the Influence of Central Hole Diameter in a Wire Mesh Electrode on Ionic Wind Characteristics

**DOI:** 10.3390/mi14081614

**Published:** 2023-08-16

**Authors:** Ji Hong Chung, Dong Kee Sohn, Han Seo Ko

**Affiliations:** 1School of Mechanical Engineering, Sungkyunkwan University, Suwon 16419, Republic of Korea; akql12@gmail.com; 2Department of Smart Fab. Technology, Sungkyunkwan University, Suwon 16419, Republic of Korea

**Keywords:** ionic wind, electrohydrodynamics, DC negative corona, central hole, needle-to-mesh configuration, energy conversion efficiency

## Abstract

Ionic wind, which is generated by a corona discharge, is a promising field that offers significant advantages by directly converting electrical energy into kinetic energy. Because of the electrical characteristics of ionic wind, most studies aiming to improve the performance of ionic wind generators have focused on modifying the geometry of electrode configurations. A mesh-type electrode is one of the electrodes used as a collecting electrode in an ionic wind generator. Using a mesh electrode results in decreased momentum of the ionic wind and increased pressure drop due to frictional loss of the flow. In this study, to minimize the reduction in momentum, a mesh electrode with a central hole was proposed and investigated. Experiments were conducted with the configuration of a needle and mesh with the central hole. These experiments analyzed the effect of the central hole diameter and the distance between the needle and the mesh electrodes on the electrical and physical characteristics of the ionic wind. The addition of the central hole led to a higher average velocity and lower current, thus resulting in increased energy conversion efficiency. The presented configuration offers a simple geometry without electrical and physical interference from complex configurations, and it is considered to have the potential to improve energy conversion efficiency and optimize ionic wind flow.

## 1. Introduction

Ionic wind—or corona wind—is induced by a corona discharge in gas at atmospheric pressure; it is a kind of non-thermal plasma. This physical phenomenon is known as electrohydrodynamic (EHD) flow. An apparatus that generates ionic wind is commonly referred to as an ionic wind generator, ionic wind device, or EHD pump. The ionic wind generator offers several advantages, including low power consumption, instant response, and the ability to generate flow on demand without having any moving mechanical parts [1,2]. Due to these benefits, ionic wind generators have attracted increasing attention in recent years for their potential applications in various fields such as flow control [3,4], heat transfer enhancement [5,6,7], microfluidics [8,9], and air cleaning [10,11].

The macro physics of ionic wind is based on the direct conversion of electrical energy into kinetic energy for fluid flow. When a large electrode such as a mesh, cylinder, or plate is connected to the positive pole and the small electrode with a sharp edge is connected to the negative pole, the electric field is mainly concentrated at the highly curved electrodes, and a negative corona discharge occurs. In the strong electric field at the curved electrode, free electrons and negative ions are generated. These charged particles drift toward the positively charged electrode. During this process, the charged particles continuously collide with neutral gas molecules; these successive collisions between charged particles and neutral gas molecules result in the transfer of momentum and the generation of ionic wind.

The performance of the ionic wind generator is influenced by various factors, including flow velocities, the presence of a dielectric, the type of discharge gas, as well as specific design variables such as electrode configuration and dimensions. The traditional configurations are needle-to-mesh [12], needle-to-ring [6,8,13,14], wire-to-rod [3,15], and wire-to-cylinder [16,17]. Recently, multiple or arrayed electrodes have been incorporated to improve the acceleration of generated flow [18,19]. Further, Moon et al. [20] proposed a complex design using a needle-to-mesh configuration with an emitting electrode surrounded by a ring electrode of the same voltage potential, which resulted in a 2.5-times increase in the energy conversion efficiency of the ionic wind generator compared to the standard needle-to-mesh configuration. These attempts showed that improved outcomes can be achieved by either increasing the overall flow rate through a larger cross-sectional area or elongating the discharge along the flow direction to create a longer interaction region. However, the more complex geometry resulting from the use of multiple electrodes can introduce additional challenges due to the electrical and physical interference between the discharge and induced flow.

Using a mesh electrode as a collecting electrode may result in higher energy conversion efficiency and velocity of ionic wind compared to those of other types of electrodes [21]. The mesh electrode causes a large pressure drop due to the frictional loss of the fluid flow, thus reducing the momentum of the ionic wind, which has prompted attempts to optimize it by changing the diameter or number of wires constituting the mesh electrode [22,23]. Despite these efforts, the fundamental structural limitation remains unsolved. It is necessary to find a way to alleviate the pressure loss caused by the mesh electrode wire entanglement structure while avoiding a complex geometry.

This study focuses on the diameter of the central hole in the mesh electrode. In the needle-to-mesh configuration, the ionic wind is the strongest in the concentric direction with the needle electrode. Creating a hole at the concentric position of the mesh electrode can suppress the pressure loss and maximize the stream of ionic wind. It is also expected that increasing the absolute distance between the emitter electrode and the mesh electrode will decrease the corona current, thereby improving the energy conversion efficiency. Further, this configuration offers a simple geometry without electrical and physical interference from complex configurations. 

The present work studied the effect of the diameter of the central hole in the mesh electrode and the distance between the needle and the mesh electrode on the generation of ionic wind. A needle-to-mesh configuration with a central hole was applied in this study. Experiments were conducted to investigate the electrical and physical characteristics of the ionic wind according to the geometrical variables. The onset voltage and corona current were analyzed to reveal the electrical characteristics of the ionic wind, and various physical characteristics induced by the central hole were discussed. The results of this study are expected to provide important insights into the design and optimization of the ionic wind generator.

## 2. Materials and Methods

Experiments were conducted to analyze the electrical and physical characteristics of the ionic wind according to the diameter of the central hole on the mesh electrode. Figure 1 shows a schematic of the experimental setup. The ionic wind generator had an inner diameter of 40 mm and was made of polycarbonate. A tungsten needle was used as a discharge electrode and a stainless steel mesh with a hole in the center made using a laser cutting method was connected to the ground of the power supply. A negative voltage was applied to the needle using a negative DC power supply (Convertech (Gyenggi-do, Republic of Korea), SHV120R-20 kV N). The voltage displayed on the power supply exhibited a difference of less than 2.5% compared to the voltage measured by the high-voltage probe (Tektronix (Beaverton, OR, USA), P6015A) and oscilloscope (Tektronix, TDS2024C). This displayed voltage was used in the calculation of the discharge power. The electric current from the mesh electrode was monitored with a multi-meter (Sanwa (Tampa, FL, USA), CD771). A hot-wire anemometer (TSI (Shoreview, MN, USA), 9565-P) was placed 25 mm away from the outlet of the ionic wind pump in x direction. It was then attached to a movable stage traveling at a velocity of 0.08 mm/s along the radial direction to measure the radial velocity distribution. To ensure the reliability of the experimental results, each experiment was repeated five times and conducted exclusively at a set temperature of (299.8 ± 1.6) K and a relative humidity of (18.3 ± 3.2)%.

The parameters in this study were the normal distance (*d*) from the needle tip to the mesh electrode and the diameter of the central hole (*D*) in the mesh electrode. Specifically, the normal distance was set to 5 mm, 10 mm, 15 mm, or 20 mm, while the diameter of the central hole was changed to 0 mm (without central hole), 5 mm, 10 mm, 15 mm, and 20 mm, respectively. With these parameters, the absolute distance between two electrodes (*L*) can be determined as follows: (1)$$L=\sqrt{d^{2}+{\left(D/2\right)}^{2}}$$

## 3. Results and Discussion

### 3.1. Electrical Characteristics

The normal distance between the emitting electrode and the collecting electrode is the main parameter of the onset voltage (*V*_0_) at which the electrical discharge is initiated [24,25]. In the configuration of this study, the absolute distance from the needle tip to the mesh electrode surface exceeds the normal distance due to the central hole in the mesh electrode. Consequently, the onset voltage shown in Figure 2 is proportional to the absolute distance as follows:(2)$$V_{0}=1.72\mathrm{Exp}\left(42.0\cdot L\right)$$

The electrical characteristics of the ionic wind generator are determined by the relationship between corona current (*I*) and applied voltage (*V*). Thus, the corona current was measured by the applied voltages in the interval of 0.5 kV. Figure 3 shows the relationship between the applied voltage and the current against the diameter of the central hole. An increase in the distance between electrodes resulted in a reduction in the measured corona current. Further, the voltage range that could be measured was narrow due to the low sparking voltage. The larger the diameter of the central hole at a given normal distance, the greater the absolute distance between electrodes, thus resulting in a lower measured corona current. Smaller normal distances exhibited a greater change in absolute distance as the diameter of the central hole increased, which resulted in a larger decrease in measured corona current.

The Townsend relation was determined from the experimental results and established as the following mathematical expression of the relationship between corona current and applied voltage [26]:(3)$$I=K_{1}V(V-V_{0})$$
where *K*_1_ is an empirical parameter that is defined by the electrode configuration and which represents the kind of sensitivity of corona current to the change in applied voltage. Figure 4 shows the Townsend relation according to the normal distances. The results indicate that stable corona discharge occurs at the tip of the needle electrode regardless of the normal distance and the diameter of the central hole.

The onset voltage, which is one of the electrical characteristics of the ionic wind generator, has a relationship with the absolute distance, *K*_1_, which represents the electric sensitivity of the configuration due to the gaseous space between the electrodes, which behaves like an air capacitor with an air dielectric. Figure 5 shows the relationship between *K*_1_ and absolute distance. A lower electric sensitivity between the corona current and applied voltage was observed as the absolute distance increased. The general relation between them could be expressed as follows:(4)$$K_{1}=m{\left(L\right)}^{n}$$
where m and n represent the functions of the diameter of the central hole, which could be expressed as 52.6 + 2170 *D*^0.763^ and 0.36 exp(−136*D*) − 2.16, respectively.

### 3.2. Flow Characteristics

The magnitude of the momentum generated by the ionic wind generator is an important characteristic to consider. The local velocity distribution was measured to analyze the momentum of the ionic wind according to the diameter of the central hole. Figure 6 plots the maximum velocity (*u*_max_) measured at the coincident location with the tip of the needle electrode according to the applied voltage at various normal distances. When the normal distance was 5 mm, the absolute distance for the plain mesh (without a central hole) was 5 mm, whereas for the mesh with a diameter of the central hole of 20 mm, the absolute distance was 11.2 mm. Thus, the corona current with a diameter of the central hole of 20 mm was less than one-third of the plain mesh, ultimately resulting in a lower maximum velocity. However, aside from a normal distance of 5 mm, the maximum velocity obtained with the same applied voltage was higher when the central hole on the mesh electrode was made, regardless of the size of the central hole and absolute distance, compared to the maximum velocity of the plain mesh. The maximum velocity increased with a decreasing diameter of the central hole, as a smaller area of the central hole resulted in more concentrated flow and larger current due to the shorter absolute distance.

Figure 7 depicts the local velocity of the ionic wind that was measured according to different normal distances when the applied voltage was 5 kV, where *r* and *R*_0_ denote the radial coordinates from the tip of the needle electrode and the inner radius of the ionic wind generator, respectively. A similar parabolic velocity distribution was obtained for each diameter of the central hole, where there were only variations in the maximum velocity. However, the parabolic curve of the plain mesh was modified with the change in normal distance. When the central hole was fabricated, the ionic wind with low momentum that was generated at a low applied voltage close to the onset voltage could pass through. Conversely, the plain mesh resulted in a loss of pressure across all areas, and the lower the momentum, the flatter parabolic curves were observed. 

With an increase in the diameter of the central hole, the normalized radius where the parabolic slope changed rapidly moved farther away from the center. It can be seen that most of the generated flow passed through the radius of the central hole. The flow can be compared with different diameters of the central holes by examining the ratio between the average velocity ($\overset{-}{u}$) at the outlet and the maximum velocity. Figure 8 shows the average ratio of each normal distance and the diameter of the central hole to all applied voltages, and the focusing of the ionic wind decreased as the diameter of the central hole increased. Moreover, the ratio decreased with an increase in the diameter of the central hole. In the ionic wind, charged particles were generated and accelerated along the electric field lines due to the Coulomb force, and the focusing of the momentum was deviated resultantly. Figure 9 shows the electric field configuration of the setup for different normal distances and diameters of the central hole, developed using the COMSOL 6.0 numerical simulation. The electric field lines formed in the radial direction around the needle tip also increased with an increase in the diameter of the central hole, therefore causing charged particles to gain more momentum in the radial direction and diffuse. Thus, when the normal distance was closer and the diameter of the central hole was larger, the ratio between the average velocity and the maximum velocity increased.

The total flow rate of the ionic wind generator can be calculated by multiplying the average velocity by the cross-sectional area of the outlet. As all the configurations in the current study have the same cross-sectional area of the outlet, the average velocity was considered to be an indicator of performance and expressed as a function of input electric power (*P* = *I∙V*). Figure 10 shows a linear relation between the cube of the average velocity and the input electrical power. The average velocity was found to be higher in all ranges when the mesh with the central hole was used than when the plain mesh was used. The slope of the relation became steeper as the diameter of the central hole increased, and a diameter of the central hole of 20 mm with a normal distance of 5 mm showed the steepest slope. The change in slope according to the normal distance was smaller for the smaller diameter of the central hole due to the difference in the change in electric field density in the radial direction, as discussed previously.

The energy conversion efficiency (*η*) is one of the most important parameters for the design of the ionic wind generator, and it can be calculated as follows [27]:(5)$$\eta =\frac{output\;mechanical\;power}{input\;electric\;power}=\frac{\frac{1}{2}\dot{m}{\bar{u}}^{2}}{IV}=\frac{\frac{1}{2}\rho {\bar{u}}^{3}A}{P}$$
where *ṁ*, *ρ*, and *A* mean the mass flow rate of the ionic wind, density of air, and cross-sectional area of outlet, respectively. Figure 11 shows the conversion efficiency calculated using Equation (5) according to the electric power. The conversion efficiency of the ionic wind generator increased when the input power rapidly increased up about 0.1 W.

However, with further increases in the input power, which varied according to the normal distance and the diameter of the central hole, the conversion efficiency tended to become gradually saturated. The highest conversion efficiency was achieved with a diameter of the central hole of 20 mm with a normal distance of 5 mm, where the slope of the cube of the average velocity was the steepest. A larger current flux resulted in a greater momentum of the ionic wind, and a smaller diameter of central holes produced larger current fluxes due to their closer absolute distances. However, a smaller diameter of the central hole resulted in a larger area of the mesh and a narrower central hole, thus leading to increased mechanical loss. As a result, the plain mesh had a significant difference in conversion efficiency compared to the mesh with the central hole, which was attributed to the absence of the central hole.

## 4. Conclusions

This study analyzed a simple mesh electrode design featuring a central hole for use in an ionic wind generator to address the structural limitations of current mesh electrodes. The experimental results revealed that the proposed needle-to-mesh configuration with a central hole enhanced the performance of the ionic wind generator compared to that without a hole. The presence of the central hole, regardless of its diameter, resulted in a lower current and higher average velocity of the ionic wind compared to the design lacking a central hole; this led to higher energy conversion efficiency. The optimal conditions for the highest conversion efficiency were a normal distance of 5 mm and a central hole diameter of 20 mm, which corresponded to the largest radial movement of the charged particle. Further, as the diameter of the central hole decreased and the distance between the electrodes increased, the ratio between the average and maximum velocity decreased as well. Although the conversion efficiency was low, the use of a smaller central hole diameter may be more effective for specific applications such as local cooling, due to the higher focusing of the momentum and maximum velocity. Through optimization of the diameter of the central hole and the normal distance for the purpose of each application, the presented configuration has the potential to significantly enhance its desired performance without the need for complex geometry.

## Figures and Tables

**Figure 1 micromachines-14-01614-f001:**
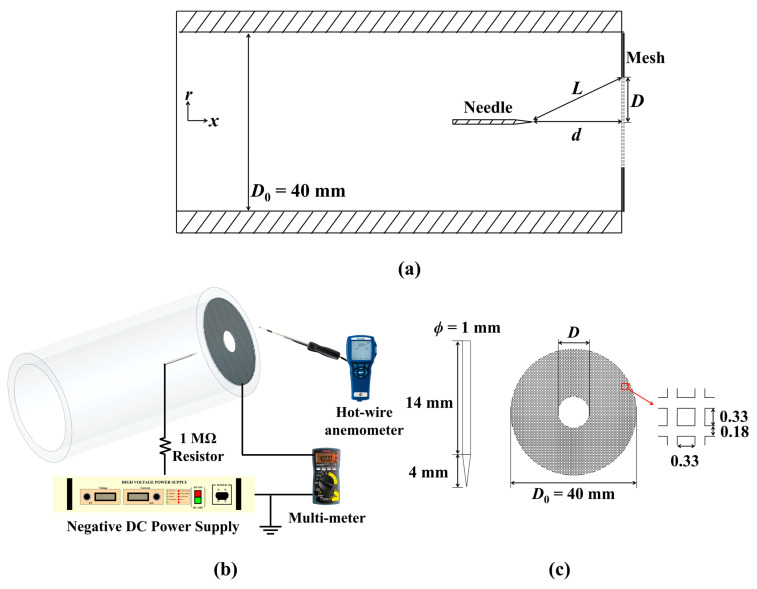
Experimental setup for analysis of ionic wind generator: (**a**) schematic (center cut of plane view), (**b**) structure and layout of device, and (**c**) dimension of needle and mesh electrode.

**Figure 2 micromachines-14-01614-f002:**
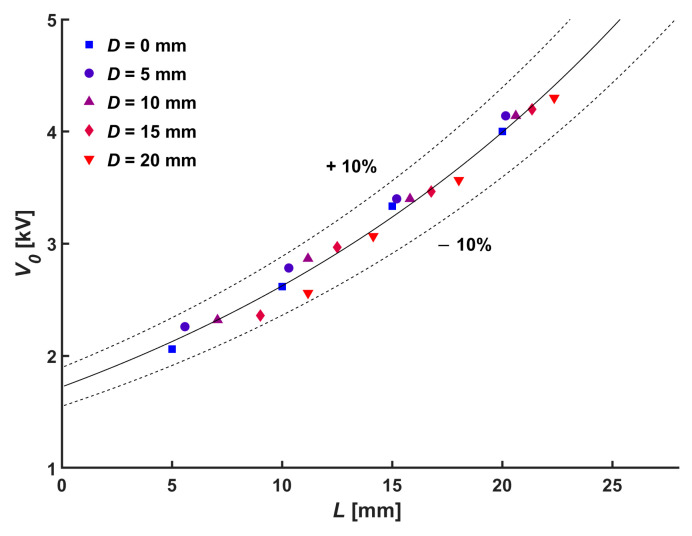
Relation between onset voltage and absolute distance.

**Figure 3 micromachines-14-01614-f003:**
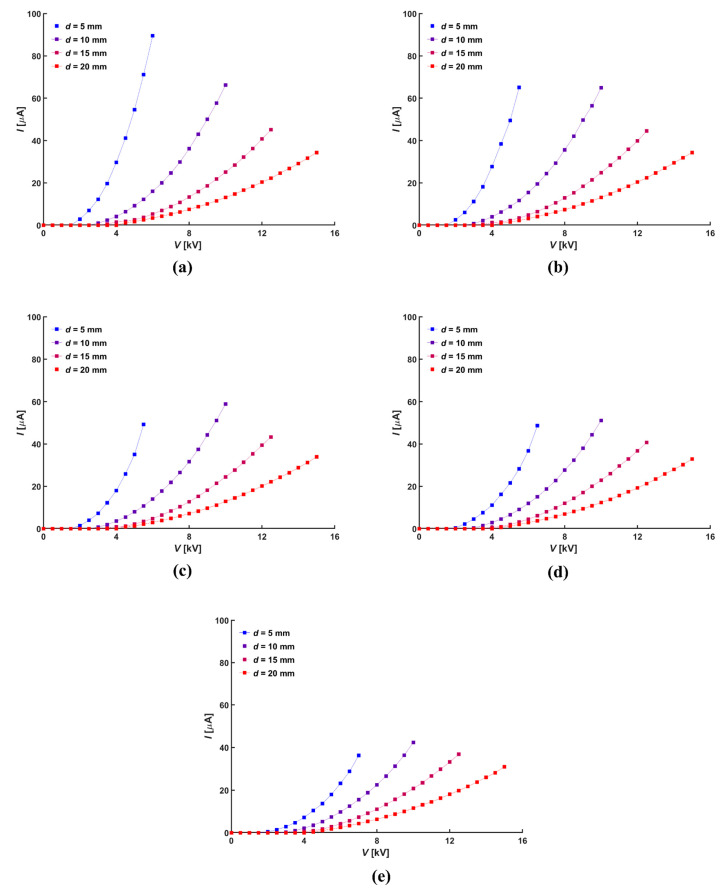
Relation between applied voltage and current according to hole diameter, *D*: (**a**) 0 mm, (**b**) 5 mm, (**c**) 10 mm, (**d**) 15 mm, and (**e**) 20 mm.

**Figure 4 micromachines-14-01614-f004:**
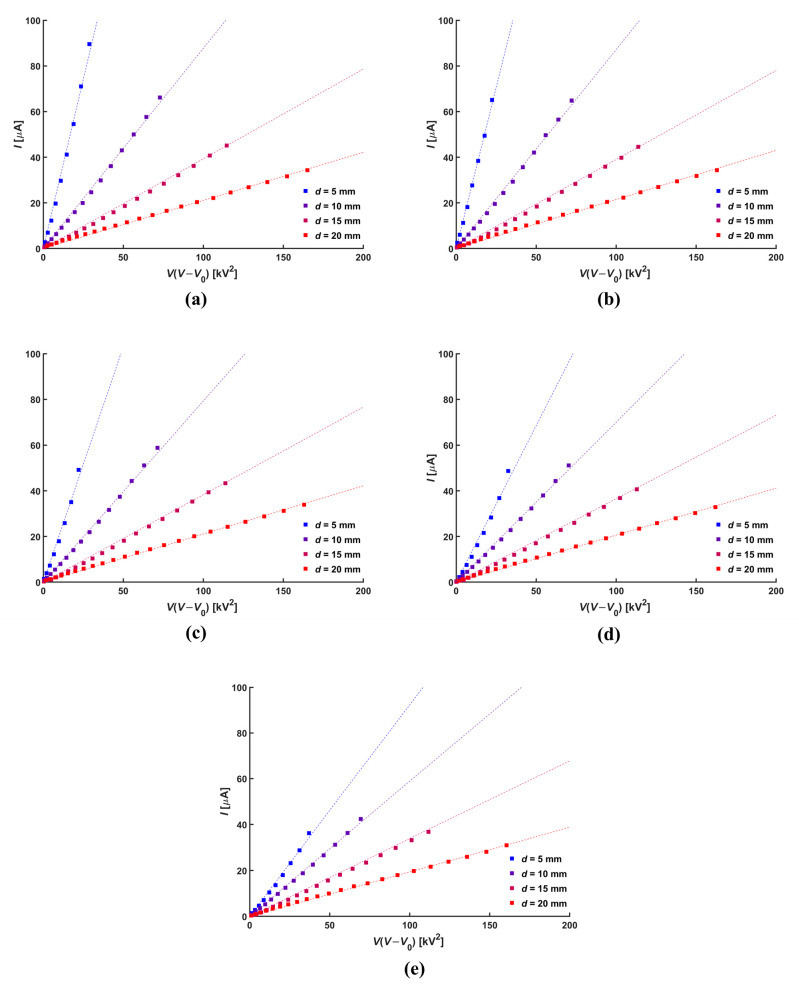
Townsend relation for various hole diameter, *D*: (**a**) 0 mm, (**b**) 5 mm, (**c**) 10 mm, (**d**) 15 mm, and (**e**) 20 mm.

**Figure 5 micromachines-14-01614-f005:**
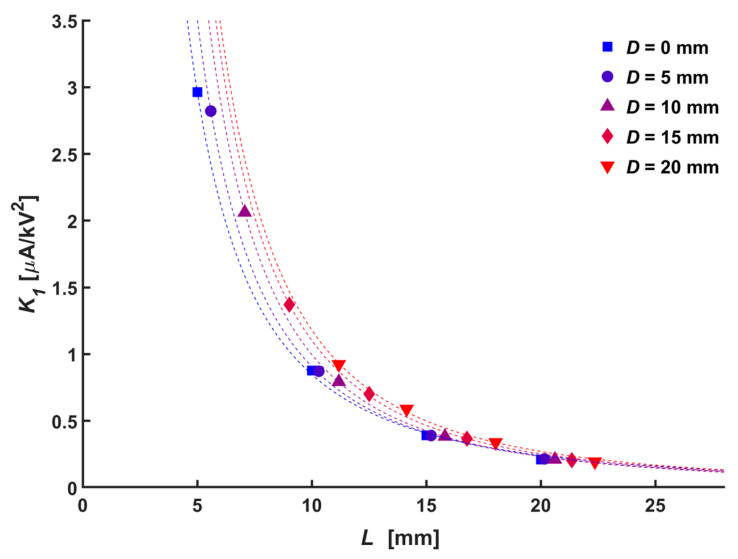
Relation between *K*_1_ and *L* according to hole diameter, *D*.

**Figure 6 micromachines-14-01614-f006:**
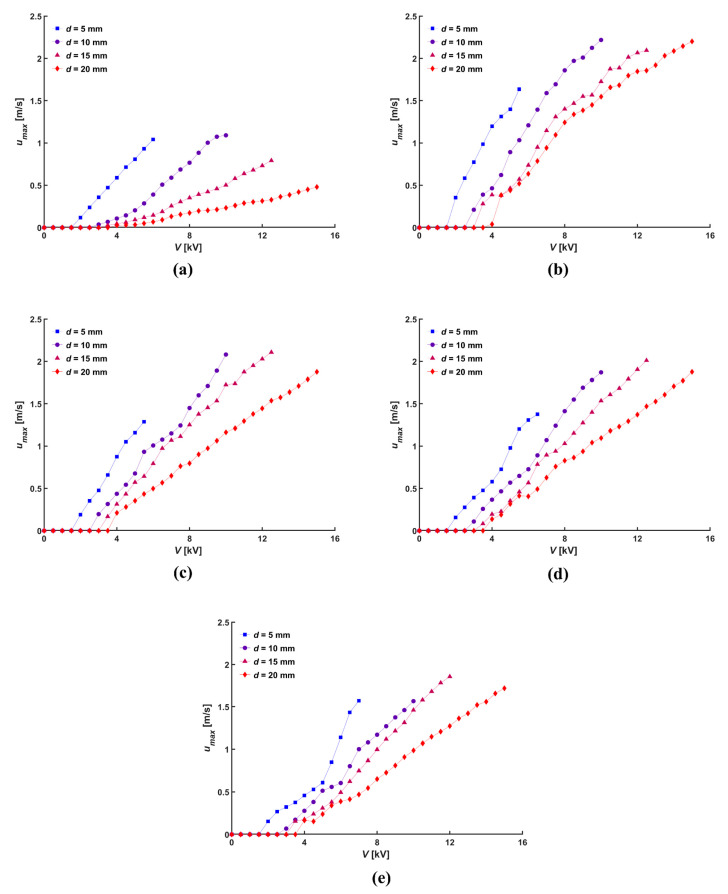
Maximum velocity of ionic wind according to various hole diameter, *D*: (**a**) 0 mm, (**b**) 5 mm, (**c**) 10 mm, (**d**) 15 mm, and (**e**) 20 mm.

**Figure 7 micromachines-14-01614-f007:**
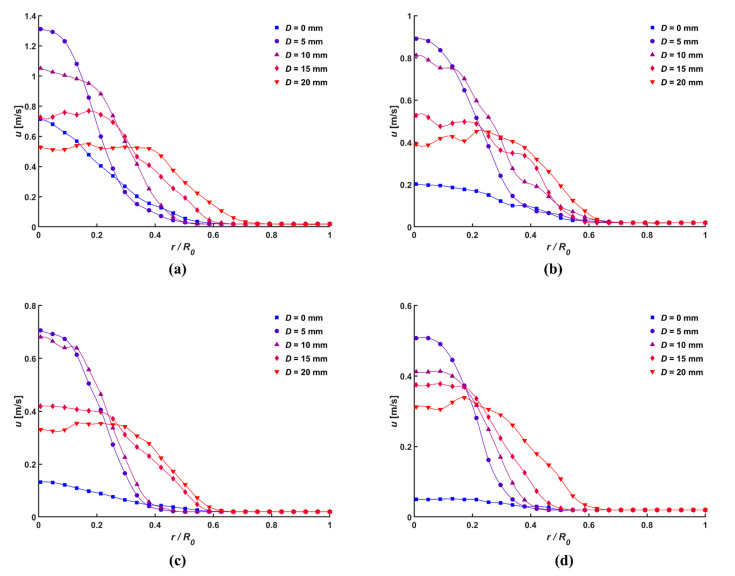
Velocity distribution of ionic wind according to various distance, *d*: (**a**) 5 mm, (**b**) 10 mm, (**c**) 15 mm, and (**d**) 20 mm, with *V* = 5 kV.

**Figure 8 micromachines-14-01614-f008:**
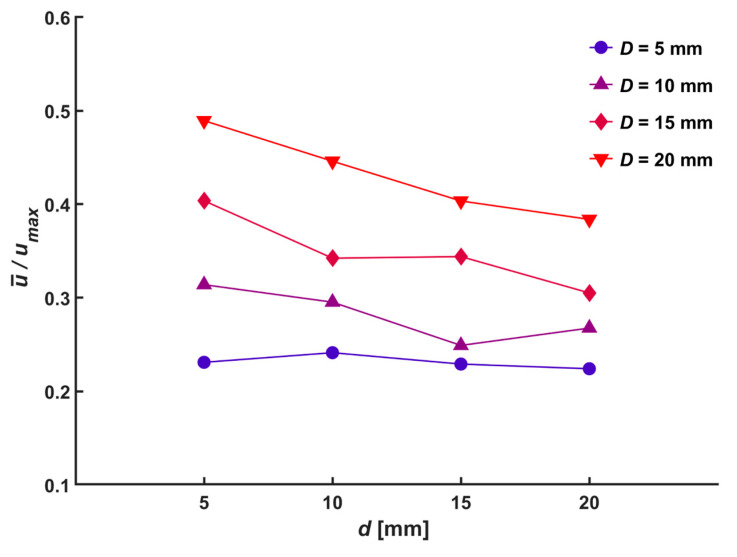
Ratio between average velocity and maximum velocity for various hole diameter, *D*.

**Figure 9 micromachines-14-01614-f009:**
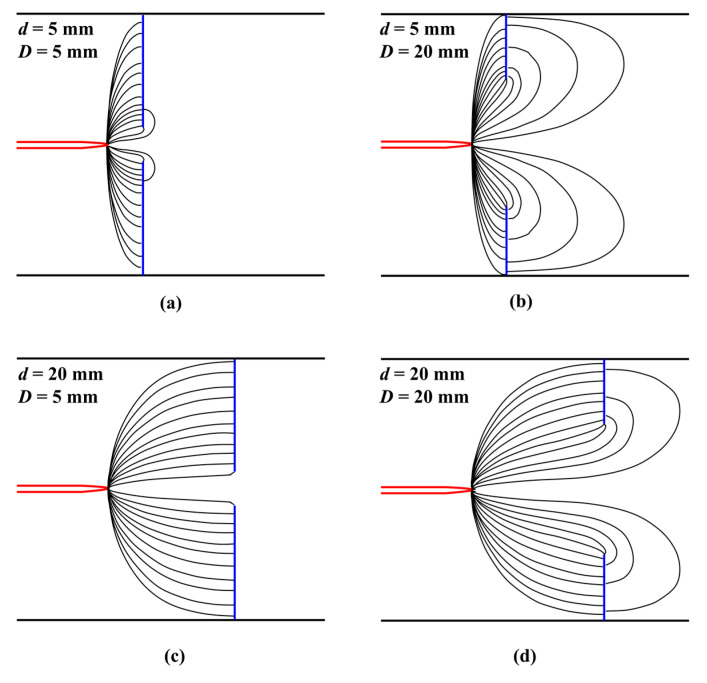
Schematic of electric field for various distance and hole diameter: (**a**) *d* and *D* = 5 mm, (**b**) *d* = 5 mm, *D* = 20 mm, (**c**) *d* = 20 mm and *D* = 5 mm, and (**d**) *d* and *D* = 20 mm.

**Figure 10 micromachines-14-01614-f010:**
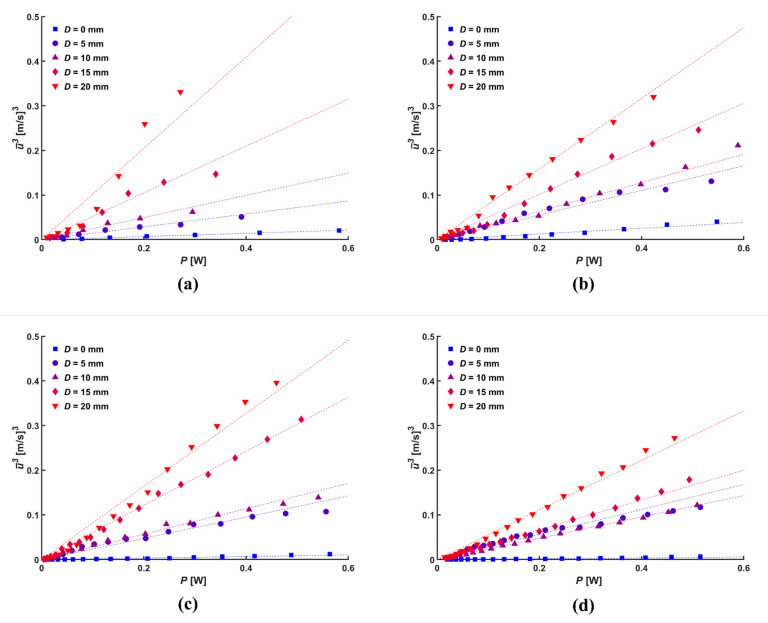
Linear relation between electric power and average velocity cubed for various *d*: (**a**) 5 mm, (**b**) 10 mm, (**c**) 15 mm, and (**d**) 20 mm.

**Figure 11 micromachines-14-01614-f011:**
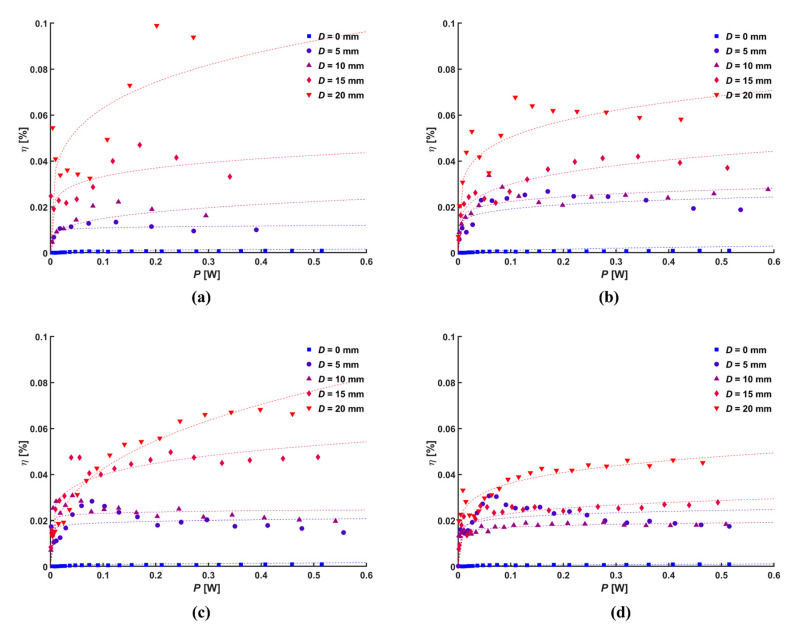
Conversion efficiency according to discharge power for different *d*: (**a**) 5 mm, (**b**) 10 mm, (**c**) 15 mm, and (**d**) 20 mm.

## Data Availability

Not applicable.

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
