# Peer review of "Study on the Influence of Central Hole Diameter in a Wire Mesh Electrode on Ionic Wind Characteristics"

_micromachines, 2023, doi:10.3390/mi14081614_

Round 1
Reviewer 1 Report
In this paper, a mesh electrode with a central hole was proposed and investigated to minimize the reduction in momentum in the ionic wind. The manuscript is well written. But I have some questions and comments:
1. The authors should add some simulations to verify the conclusion.
2. It is coarse to calculate the output mechanical power based on the measured velocities if the authors cannot provide detailed velocity distributions.
Reviewer 2 Report
The manuscript "Study on Effect of Hole Structure in Mesh Electrode on Ionic Wind Characteristics" presents experimental studies of a new corona discharge system for the production of ionic wind. In the tested system, needle-grid corona discharges were used. Note that this is a needle configuration with a perforated metal electrode with square holes (the real mesh is made of thin wires that form a corrugated structure, in the case described it is a flat perforated metal). The suggested title of the manuscript seems to be, for example, "Study on the influence of the diameter of the hole made in the perforated metal electrode on the characteristics of the ionic wind in corona discharges".
1) The manuscript describes that the Sanwa CD771 multimeter was used to measure the discharge current (minimum range 400μA, measurement error 1.3 μA), it should be noted that the discharge currents are initial values of the order of fractions of μA, an oscilloscope is used to measure such low currents because the use of a multimeter is burdened with a large measurement error.
2) In the abstract, the authors wrote "
A mesh-type electrode is commonly used as a collecting electrode in an ionic wind generator. " Please provide a publication that supports this statement.
3) The authors did not mention the edge effect that occurs in the case of holes, edges or needles, they did not present, for example, photos showing in which area discharges occur on a perforated electrode.
4) The authors write in the introduction "
The ionic wind generator offers several advantages, including low power consumption, silent operation, instant response, and the ability to generate flow on demand without having any moving mechanical parts ". This is not entirely true because cracks are generated during the discharge of the system, which generates sound and is therefore not silent.
5) In the following paragraphs, the authors write " The performance of the ionic wind generator is significantly affected by certain design variables, such as electrode configuration and dimensions. " The authors are wrong because discharges are influenced by many more factors, among others; frequency, flow velocities, electrode distance, dielectric presence, gas type and other factors not mentioned by the authors.
6) In the following chapter, the authors wrote " The onset voltage, which is one of the electrical characteristics of the ionic wind generator, has a relationship with the absolute distance, K1, which represents the electric sensitivity of the configuration due to the gaseous space between the electrodes, which be haves similar to resistance in an electric circuit system” This is not entirely true, it behaves like an air capacitor with an air dielectric.
7) Figure 9 shows the configuration of the electric field for different normal distances and diameters of the central hole. I have a question on what basis were the drawings developed, since the distribution of the electric field was not measured, and simulations were not carried out?
8) An important factor affecting the generation of ion wind is the influence of temperature, which reduces the efficiency of the system. The manuscript does not say anything about the temperature that affects the reduction of ionization (in corona discharge the temperature is higher than in DBD or SD systems)
9) The description of the manuscript does not show which measuring equipment was used to measure the high voltage because the Sanwa CD771 meter does not support the 4kV range
10) The frequency of generating pulses and the method of frequency measurement are not described because the influence of frequency is also important when expressing the discharge power.
The manuscript contains a large number of shortcomings concerning the descriptions, but above all errors related to the measurements of the equipment used to carry out the measurements. The meters used are not adapted to carry out measurements in the described range or their accuracy is low for experimental research, the results obtained are burdened with a large error, which makes their scientific usefulness troublesome and easy to undermine due to the measuring equipment used, e.g. currents and voltages. In order to introduce the appropriate corrections, the manuscript should be sent for re-review, the graphs and the received data will certainly be changed
Some phrases require correction, but this does not affect the content of the manuscript.
Round 2
Reviewer 2 Report
The authors introduced corrections in the manuscript, however, the measurement of the current using the Sanwa CD771 multimeter discussed by the authors is not very precise, the authors in response sent a graph of the measurement using the P6015A courts and the TDS2024C oscilloscope.
1) Please send a diagram of the current measuring system using an oscilloscope, how the discharge current was measured using the elements; judgments of the P6015A and the TDS2024C oscilloscope on how the current was calculated using the oscilloscope.
2) In the body of the article, the authors wrote "To address this, we ensured that the sharp edge of the central hole faced the opposite direction of the needle electrode in all subsequent experiments, thus ensuring better reproducibility in our experiments". In the drawing, the sharp edges are visible and it is not possible for all the edges to be bent/deviated in the opposite direction because it would require bending the wires, which would change the geometry of the hole, which would lose its oval shape, which passes the direction of the tests, so the edge effect is inevitable.
3) The authors still have not explained the issue of the origin of Figure 9 in the manuscript. As the authors write in their answers to questions, Figure 9 was developed using the COMSOL numerical simulation. The above-mentioned information is not in the manuscript, which may make it difficult to understand on what basis it was made drawing for potential reader
4) Question 3 was about the effect of the discharge temperature, not the ambient temperature, because the ionic wind is generated in the surroundings of the discharge and the excessive temperature of the discharge affects the generation of ions.
5) Calculation of the discharge power based on the formula (P = U * I) is a major simplification because it does not take into account the frequency of the pulses because the discharge power will be different when there is, for example, 1 pulse per second lasting 100ms and the discharge power is different when it is 3kHz. The influence of the frequency on the obtained power significantly affects the calculated efficiency.
6) There is no information in the manuscript what type of pulses are used (rectangular, sawtooth) and what was the frequency of the pulses The efficiency values calculated by the authors are extremely low and constitute hundredths of a percent (0.6 per mille), which is almost a measurement error.
These issues should be corrected and included in the manuscript by the authors so that they do not raise doubts and do not pose problems for the reader with understanding the experiment and the credibility of the presented research results.
The manuscript contains minor linguistic errors that do not affect the content.
Round 3
Reviewer 2 Report
The authors answered the question and provided figures that clarify the question.